# Methods for Inferring Cell Cycle Parameters Using Thymidine Analogues

**DOI:** 10.3390/biology12060885

**Published:** 2023-06-20

**Authors:** Joaquín Martí-Clúa

**Affiliations:** Unidad de Citología e Histología, Departament de Biologia Cel·lular, de Fisiologia i d’Immunologia, Facultad de Biociencias, Institut de Neurociències, Universidad Autónoma de Barcelona, Bellaterra, 08193 Barcelona, Spain; joaquim.marti.clua@uab.es

**Keywords:** thymidine analogues, S-phase, cell cycle parameters, cell kinetics, toxicity

## Abstract

**Simple Summary:**

Several procedures have been developed to infer cell-cycle time and the duration of the cycle phases, including tritiated thymidine autoradiography and the immunodetection of thymidine analogues after their incorporation into replicating DNA. These methods have supplied important insights to reveal, on fixed tissue sections and using light or electron microscopy, the proliferative behavior of different cell types under different experimental contexts. Thymidine analogous labeling has provided knowledge about cell growth kinetics that would never have been obtained by histological methods alone.

**Abstract:**

Tritiated thymidine autoradiography, 5-bromo-2′-deoxyuridine (BrdU) 5-chloro-2′-deoxyuridine (CldU), 5-iodo-2′-deoxyuridine (IdU), and 5-ethynyl-2′-deoxyiridine (EdU) labeling have been used for identifying the fraction of cells undergoing the S-phase of the cell cycle and to follow the fate of these cells during the embryonic, perinatal, and adult life in several species of vertebrate. In this current review, I will discuss the dosage and times of exposition to the aforementioned thymidine analogues to label most of the cells undergoing the S-phase of the cell cycle. I will also show how to infer, in an asynchronous cell population, the duration of the G_1_, S, and G_2_ phases, as well as the growth fraction and the span of the whole cell cycle on the base of some labeling schemes involving a single administration, continuous nucleotide analogue delivery, and double labeling with two thymidine analogues. In this context, the choice of the optimal dose of BrdU, CldU, IdU, and EdU to label S-phase cells is a pivotal aspect to produce neither cytotoxic effects nor alter cell cycle progression. I hope that the information presented in this review can be of use as a reference for researchers involved in the genesis of tissues and organs.

## 1. Introduction

The cell cycle is the name given to the orderly sequence of events by which a cell passes its genetic material to its progeny. In eukaryotic cells, this essential biological process is separated into two main events: interphase and mitotic phase. In the first of these, the cell grows and duplicates its DNA. This event takes place in three phases: gap 1 (G_1_), DNA synthesis (S-phase), and gap 2 (G_2_) [1]. The mitotic phase (M), on the other hand, is divided into five well-defined phases: prophase, prometaphase, metaphase, anaphase, and telophase [2] (Figure 1). During these phases, the cell separates its DNA into two sets. Cytokinesis is the final physical cell division that follows telophase. In this step, the cell divides its cytoplasm, generating two identical daughter cells [3,4]. After finishing the cycle, a cell can either start a new cell cycle or enter a resting state called G_0_. From this phase, a cell can undergo terminal differentiation [5] (Figure 1). The progression of the cells through the cell cycle is strictly controlled by highly orchestrated steps reacting to intracellular and extracellular signals [6].

Various approaches have been used to identify the phases of the cell cycle. One of these is based on the analysis of DNA content using flow cytometry. The cells in G_1_ and G_0_ have half the DNA content (2N DNA content) that G_2_ and M cells have (4N DNA content) [1,7]. In flow cytometric analysis, the most frequently used protocols for DNA staining of fixed cells are based on the use of stoichiometrically dyes, which bind to double-stranded DNA. There are three examples of these: propidium iodide, 4,6-diamidino-2-phenylindole, and Hoechst 33258 or Hoechst 33342 [6].

Alternatively, to infer the fraction of S-phase cells and determine cell cycle parameters and phase durations, several strategies have been applied for the detection of modified nucleotides incorporated into replicating DNA. The original procedures to detect S-phase cells involved the use of tritiated thymidine ([^3^H]TdR) and a detection step using either high-resolution autoradiography or scintillation techniques [8,9]. For decades, [^3^H]TdR and autoradiography have produced critical insights into the cellular mechanisms of cell proliferation. However, because this procedure is expensive, requires technical expertise in handling radio-labeled products, and the procedure takes several weeks in developing autoradiographs [10,11], contemporary studies are carried out with thymidine analogues. These studies have been facilitated by advances in the production of monoclonal antibodies against thymidine analogues [12].

The thymidine analogues are modified in the 5 position of the thymidine ring by halogen atoms (bromine, chlorine, and iodine). Of all of them, 5-bromo-2′-deoxyuridine (BrdU) is one of the most frequently used to infer duration and phases of the cell cycle [6]. 5-chloro-2′-deoxyuridine (CldU) and 5-iodo-2′-deoxyuridine (IdU) are halogenated thymidine analogues that resemble BrdU in their ability to label duplicating DNA. They have allowed us to perform studies involving double and triple S-phase labeling schemes [13].

BrdU, CldU, and IdU detection needs the production of single-stranded DNA because antibodies cannot gain access to native DNA [12,14]. Usually, this methodological condition is obtained by incubating the tissue sections in hydrochloric acid [15]. Pretreatment with this antigen retrieval may erode cell and tissue components [13,16]. To bypass these negative effects, a modified analog of thymidine, 5-ethynyl-2′-deoxyiridine (EdU), was used to label replicating DNA because its detection is performed with the click chemistry reaction [17].

This current review will show how to reveal the fraction of S-phase cells and how to infer cell cycle parameters and phase durations in an asynchronous cell population (the cells in the population are randomly distributed throughout all phases of the cell cycle) because some synchronization methods (mechanical isolation of cells at specific phase of the cell cycle and the treatment of asynchronous cell populations using chemical agents) are linked to deleterious effects on the proliferative behavior of cell precursors [18,19]. For example, when hydroxyurea (a ribonucleotide reductase inhibitor) is added to a culture medium in which cells are growing, the synchronization procedure by this agent leads to cellular arrest at the G_1_/S boundary [18]. Moreover, HU selectively kills cells in the S-phase [20,21].

The following items will be addressed: (i) thymidine analogues and cell toxicity, (ii) thymidine analogues and saturation of the S-phase, (iii) inferring the cell cycle time after a single administration of thymidine analogues, (iv) estimating duration and phases of the cell cycle after cumulative labeling with thymidine analogues, and (v) inferring the cell cycle time and duration of the synthetic phase after double labeling.

## 2. Thymidine Analogues and Cell Toxicity

The thymidine analogue labeling is a powerful tool used to reveal the fraction of S-phase cells, and to infer duration and phases of the cell cycle. Despite this, the results obtained with thymidine analogues should be interpreted with caution because they present unforeseen problems. This is because BrdU, IdU, CldU, and EdU have a different chemical structure in comparison to endogenous thymidine. The methyl group at the 5 position of the thymidine ring is replaced by the halogen atoms bromide, chlorine, and iodine. EdU, on the other hand, presents a terminal alkyne group in the same 5 position [22]. When administered, they are integrated as a totally foreign atom into replicating DNA, generating important changes in the double helical structure of this nucleic acid [23]. From these data, it is reasonable to presume that the genes that use this modified DNA are unlikely to transcribe appropriately into RNA and, eventually, the proper protein [11].

It has been reported that both CldU and IdU alter cell cycle progression on tumoral cells [24,25]. No data are available in normal conditions. Therefore, there are no data available to know if these thymidine analogues present more or less toxicity than BrdU [13]. EdU administration, on the other hand, is accompanied by the deformations of the cell cycle, the slowdown of the S-phase, and the induction of interstrand crosslinks [6]. EdU is more toxic than BrdU [6,13,22].

BrdU is the most used thymidine analogue. Several studies have demonstrated that BrdU presents a myriad of detrimental effects both in vitro and in vivo [11,26,27,28,29]. They should not be ignored. In the context of the cell cycle studies, it has been revealed that when cell lines derived from murine embryonic stem cells are exposed to BrdU, they lose the expression of stem cell markers such as Nestin, Sox2, and Pax 6, and undergo glial differentiation, upregulating the astrocytic marker GFAP. The latter was paralleled by a reduced expression of DNA methyltransferases and a decrease of DNA methylation, suggesting that BrdU-tagged embryonic stem cells alter their DNA methylation status [30]. Other authors have reported that a single, low dose of BrdU has a severe antiproliferative effect in cultured neural stem cells, which is accompanied by altered cell differentiation, cell phenotype, and protein expression consistent with the induction of senescence [31]. In vivo studies, on the other hand, have shown that a single administration of BrdU at doses ranging from 100 to 300 mg/kg is able to alter the proliferative behavior of neuroblasts and leads to the activation of apoptotic cellular events in the rat cerebellar neuroepithelium [21]. In line with this, it has been shown that BrdU is an antitumoral agent because a single exposure to this thymidine analogue produces a severe impairment of cell cycle (accumulation in the G1 phase) as well as a lengthening of the time that the cells remain in S-phase [31,32]. Moreover, it is essential to accept that the detection of BrdU, IdU, CldU, and EdU, in tissue sections, should not be a demonstration that this cell was in the S-phase of the cell cycle during marker administration. This is because other conditions, such as DNA repair, cell differentiation, abortive cell cycle, and gene duplication, also promote BrdU, IdU, CldU, and EdU labeling [11,26,27].

## 3. Saturating S-Phase

An important issue when inferring the cell cycle parameters and phase durations is that related to the saturating dose, which is defined as the dose of a given thymidine analogue necessary to label most cells undergoing the S-phase of the cell cycle [13]. This parameter is necessary to have a reliable picture of the cells engaged in DNA synthesis during the exposure to the marker [33,34], which will supply accurate data and foster the comparison of results across research groups. In this context, taking the thymidine analogue BrdU as an example, it is important to have a guarantee that alterations in the fraction of BrdU-stained cells originate from variations in the proliferative status, not from variations in the methodologies used to the detection of BrdU-positive cells. In this line, an important step in BrdU immunohistochemistry is the production of single-stranded DNA to make the incorporated BrdU accessible to the antibodies. It has been demonstrated that the detection of BrdU-positive cells is affected both by the distinct antigen retrieval procedures used in such detection [14] and by the anti-BrdU antibody used [35].

To evaluate the saturating dose of BrdU, animals separated into several experimental groups are administered with a single intraperitoneal injection of this thymidine analogue, and they are sacrificed at regular intervals of time (usually from 0.5 to 3 h). An example is supplied in Table 1.

To determine the saturating dose during the prenatal development of the nervous system, the doses of BrdU tested are usually in the range of 35 to 50 mg/kg body weight. These doses have been selected because they present neither prominent cytotoxic effects on the proliferative behavior of cells nor differences in BrdU-signal, which allow for a confident identification of those cells engaged in DNA synthesis [21].

The determination of a saturating dose of BrdU has been reported for proliferating cells in the cerebellar neuroepithelium of rats. In these experiments, groups of dams were administered with a dose of BrdU (35 or 50 mg/kg body weight) at prenatal life and their offspring were allowed to survive for 0.5, 1, 1.5, and 2 h after marker exposure [34]. In each survival time, the labelling index (LI) was calculated as a percentage of BrdU-positive interphase nuclei per total number of scored neuroepithelial cells, i.e., the fraction of cells immune-labeled with BrdU at any time [14]. Typically, the LI presents an initial rise, in which only a small fraction of cells appeared labeled. It is followed by a plateau where the survival time does not increase the fraction of labeled nuclei. The survival time where the LI achieves stable values indicates the saturating dose. As an example, see Figure 3 in [34]. It was observed from that figure, that the LI presented a plateau (46%) from 1 h on after BrdU administration. These results have suggested that LI after a one-hour period provides an adequate estimation of the proportion of neuroepithelial cells in the S-phase. To my knowledge, the saturating dose of 5-chloro-2′-deoxyuridine, 5-iodo-2′-deoxyuridine and 5-ethynyl-2′-deoxyuridine have not yet been determined.

When the saturating dose of BrdU is compared among laboratories during the development of the nervous system, the data are controversial. This is because Takahashi et al. [36] have reported, in the cerebral wall of the mouse embryos, that a single administration of BrdU (50 mg/kg body weight) into pregnant dams is related with the labeling of 100% S-phase nuclei over an interval extending from 15 min to 2 h. The reasons for these divergences are unknown, but factors such as assay sensitivity and the kinetics of growing cell populations may be involved in the optical microscopic detection of BrdU. Moreover, these discrepancies have also emphasized that the saturating dose of BrdU should be prudently interpreted because, as indicated [13], this parameter depends on the species used, the organ analyzed, and the life stage studied.

Similarly, other studies [14,33] focused on the perinatal life reported determination by quantification of the LI and of the saturating dose in the proliferative zone of the rat cerebellar external granular layer after a single administration of BrdU (50 mg/kg body weight) at postnatal days 4 and 10. This dose was found to enable labeling of 31.2% of the proliferating neuroblasts after 1 h of BrdU administration, indicating that a saturation level was reached from this moment on. When this saturating dose, determined with BrdU, is compared with those saturating doses in the literature based on [^3^H]TdR administration, it is observed that LI value obtained with BrdU (31.2%) is different in comparison with those LIs reported from rodents sacrificed 1 h after administration of [^3^H]TdR. For instance, in 6-day-old rats killed 1 h after injection of [^3^H]TdR, the LI values of 27.2 [37] and 24.7% [38] were reported, but the LI in 2-day-old mice was of 45% [39], while values of 50 and 43% were determined in mice collected at postnatal days 7 [40] and 10 [38], respectively. The reason for these discrepancies are not clear, but differences between the incorporation of BrdU and [^3^H]TdR into DNA should be considered. In this context, several investigations have revealed the lack of BrdU uptake in the early S-phase [41,42]. On the other hand, these differences might be related to the different regions of the external granular layer studied. This is because even in the region of the vermis, the LI is different in the anterior versus the posterior lobe [43].

## 4. Estimating the CELL Cycle Time after a Single Administration of a Given Thymidine Analogue

In this labeling scheme, groups of rats are administered with a single dose of a given thymidine analogue and they are sacrificed following a wide range of survival times after marker administration. As an example, Table 2 depicts the schedule of BrdU administration and the survival times after the injection of this marker.

The premise of this labeling scheme is based on the assumption that the thymidine analogue will be incorporated by those proliferating cells engaged in DNA synthesis during marker supply. To infer generation times of cells in fixed tissue sections following this labeling scheme, it is necessary to bear in mind that if synchronization procedures are not used, the incorporation of a given analogue into replicating DNA will label a cohort of asynchronous cycling cells in the synthetic phase of the cell cycle. The posterior course of division cycle can be quantitatively studied from the rhythmic appearance and disappearance of labeled mitotic cells, i.e., the variation that takes place in the proportion of labeled mitosis (PLM) [34]. This assay is based on the ability to distinguish mitotic cells microscopically by their characteristic morphology [44]. The PML is defined as the percentage of labeled mitotic figures per total number of mitotic cells. A labeled mitosis is defined as a mitotic figure with either all or part of each individual chromosome labeled [33]. Examples of unlabeled and labeled mitotic cells are shown in Figure 2.

The duration and phases of the cell cycle are inferred from the graphic representation of the PLM plotted as a function of survival time after marker administration, which describes a sinusoidal curve. This curve can be generated using a polynomial regression model of fifth order. The precision of this method depends on the length of intervals between the sample collections [6] (Figure 3). At this point, it is necessary to indicate that two comprehensive analyses can be carried out. They will be named “short procedure” and “long procedure” and describe two different sinusoidal curves. In both procedures, a marker is given according to two schedules where both interjection intervals and marker doses are similar. As an example of the “short procedure”, see Figure 1 in [45], Figure 4 in [33], Figure 8a–b in [46], and Figure 4 in [47]. For the “long procedure”, see Figure 3 in [48], Figures 2–4 in [49], Figure 4 in [50], Figure 8 in [51], and Figure 4 in [34].

It is observed, in both labeling schemes, that as the time of marker exposure increases, the PML starts growing until they reach a peak. Thereafter, this percentage starts decreasing and later, in the “long procedure”, the PML begins to increase again. A second peak is observed but, depending on the study [50], the sinusoidal curve does not show a regular smooth repeating pattern. This is because the second peak is usually broader and of lower amplitude than the first peak. Several reasons can explain these differences, including variations in the proliferative status of individual cells, which will be more remarkable as the number of cycles increases, and that an unknown number of cells may stop cycling after the first peak [50].

From the sinusoidal curves described by the “short and long procedures”, a mean duration of the second gap phase or G_2_ phase (T_G2_, post-DNA synthetic phase) + 1/2 mitotic phase (T_M_), T_G2_ + 1/2 T_M_, can be inferred. It is equal to the time required for 50% of the mitotic figures to become BrdU-labeled. The duration for the S-phase (T_S_, DNA synthetic phase), on the other hand, is calculated from the elapsed time between the 50% intercepts of the ascending and descending limbs [33,48]. In addition to that, from the sinusoidal curve obtained with the “long procedure”, the duration of the whole cycle (Tc) can be determined. It is the time necessary to pass from the first 50% of tagged mitosis to the third. The duration of G1 (pre-DNA synthetic phase) + 1/2 M (T_G1_ + 1/2 T_M_) was then calculated by subtracting T_G2_ + 1/2 T_M_ + T_S_ from Tc. On the other hand, T_M_ is found by the equation: T_M_ = MI/100 × Tc/GF. The mitotic index (MI) is defined as a percentage of mitotic figures per total number of cells. The growth fraction (GF), defined as the percentage of dividing cells to the whole population in study, was calculated from the following equation: GF = Tc/T_S_ × LI/100 [14,34,48].

Estimating the cell cycle time after a single administration of a given marker presents advantages and disadvantages. The advantages are that the PLM enables measurement of the duration and phases of the cell cycle, and this procedure is insensible to the presence of senescent or quiescent cells [6]. In contrast, the disadvantages are that type of labeling indicate do not indicate additional rounds of division or the exit of the cell cycle soon after having been tagged [13].

## 5. Inferring Duration and Phases of the Cell Cycle after Cumulative Labeling

This scheme of labeling has been extensively used for inferring cell cycle parameters. It establishes that injections of a given thymidine analogue spaced at intervals of time no greater than the length of the S-phase will assure continuous saturation labeling of S-phase cells [33,34,51]. Table 3 depicts an example of cumulative labeling with BrdU and the survival times after the injection of this marker.

To apply this labeling method, it should be assumed that cells in the population are randomly distributed throughout all cell cycle phases. Moreover, to perform this labeling procedure, the choice of time intervals between the repeated administration of the marker should be carefully planned. In this context, preliminary estimation of the S-phase length is necessary [13].

To determine generation times of cells in fixed tissue sections following the cumulative labeling, several injections of a marker are given to groups of animals at successive intervals, and they are periodically sacrificed for determining the LI. The most remarkable feature of this procedure is the increasing number of labeled nuclei, i.e., the LI will increase linearly until all proliferating cells have been labeled. The increase occurs because some cells have left S-phase. They will remain labeled, while other cells have entered S-phase. Therefore, they will also be labeled [34,51,52,53]. After that, stable values are achieved, indicating that all cells moving in the cell cycle have been tagged (Figure 4).

Two important aspects of this labeling method should be considered. The first of them is the LI value. If this parameter is less than 100%, it should be assumed that the analyzed cell population presents non-cycling cells as well [6]. The second aspect of this labeling method is related with the linear increase of the LI. As mentioned, [13] if the increase is not lineal, the studied proliferative cell population is neither homogeneous (more than one proliferative cell population is present) nor asynchronous.

Adopting a rationale which has been applied under conditions of cumulative labeling [54,55,56], values for T_S_, GF, and Tc can be inferred from the y-intercept (=T_S_/Tc × GF) and from the time at which the curve approaches the asymptote—in other words, the time required to tag the complete GF. The latter represents the extent of Tc–Ts and the GF corresponds to the maximum LI attained at Tc–T_S_. From the least-squares best-fit curve, T_S_ and Tc can be estimated.

In a different approach for this labeling method, cells are labeled with EdU for gradually increased times, and the signal intensity is then studied. Maximal EdU-coupled fluorescence intensity is reached if pulsing times match the length of the S-phase [6,46]. Similarly, the duration and phase of the cell cycle can be inferred by cumulative labeling combined with the detection of the nuclear protein Ki-67 [53].

## 6. Estimating the Cell Cycle Time and Duration of the Synthetic Phase after Double Labeling

This sequential procedure enables the estimation of the length of the cell cycle. This method is based on the use of two thymidine analogues delivered within a known time interval [6,13]. The markers used may be IdU and CldU [57], BrdU and EdU [58,59], EdU and BrdU [60], IdU and BrdU [61,62], BrdU and [^3^H]TdR [63], or [^3^H]TdR and ^14^C-Thymidine [64]. The main assumptions of this procedure are: (i) the proliferating cells are assumed to be in a random phase of the cell cycle, (ii) the cell cycle length is constant, and (iii) the cell population will not increase in size during the exposure time to the markers. The disadvantage of this arrangement is that it includes quiescent and dead cells [6].

A point that deserves attention is the immunodetection, in the same tissue section, of BrdU and CldU, BrdU and IdU, BrdU and EdU, or IdU and CldU. Because of the high similarity among these thymidine analogues, it has been indicated that most primary anti-BrdU antibodies have cross-reactions to IdU, CldU, and EdU [65,66,67,68]. These unwanted cross-reactivities may complicate the multiple labeling of cells. These limitations can be overcome with the use of a specific anti-BrdU antibody derived from a specific cell line. For instance, the mouse monoclonal anti-BrdU antibody (clone MoBU-1) has no cross-reactivity with EdU. On the other hand, the mouse monoclonal anti-BrdU antibody (clone B44), which is sensitive to BrdU and IdU, presents reduced cross-reactivity to CldU, whereas the rat monoclonal anti-BrdU antibody (clone BU1/75) reacts with both BrdU and CldU, but it is insensitive to IdU [13].

There are different approaches for the estimation of the cell cycle time and duration of the synthetic phase following double labeling. An example of this method is: animals receive an initial administration of BrdU and, for instance, 2 h later, they receive a second injection of EdU. The animals are sacrificed, for instance, 1 h later. All cells of the cell kind analyzed are in the synthetic phase of the cell cycle at the time of the first injection and become labeled with BrdU. During the time interval between the two injections, the tagged cells at the end of the S-phase leave this phase and enter G_2_. On the other hand, untagged G_1_ cells enter the S-phase during the interval between the two injections. All cells that are in S-phase at the time of the EdU administration become labeled. Consequently, three cohorts of differently-labeled cells after double labeling exist: (i) labeled cells with BrdU that have left the S-phase between the administration of both markers, (ii) tagged cells with EdU that have entered S-phase between the two administrations, and (iii) double-labeled cells with BrdU and EdU that were in the S-phase at both times of injections [6,64].

Several approaches to this procedure are used for inferring the Tc:
This approach has been proposed by Hwang et al. [60]. In this case, EdU administration is followed by the BrdU administration after a known time interval. The rationale of this approach is that during the EdU administration, some cells leave the S-phase. This fraction of cells is identified by the absence of the BrdU signal. The following equation is used for inferring Tc:
(1)$$Tc=\frac{It}{P}$$
where “Tc” is the length of the cell cycle, “It” is the time period between EdU and BrdU labeling, and “P” is the proportion of cells labeled only by EdU.

2.This procedure has been proposed by Martynova et al. [69]. In this case, pregnant dams were injected with IdU and then 1.5 h later to BrdU. Animals were sacrificed after 30 min. The rationale of this method is the proportion of cells tagged by the IdU administration as well as all cells labeled by the second administration served for estimating the T_S_. The length of this phase is based on the presupposition that the ratio between both fractions of cells is equal to the ratio between the length of the time interval between both labeling pulses and the T_S_. From the following equation, T_S_ can be inferred.

(2)$$\mathrm{Ts}=\mathrm{It}\times \frac{\mathrm{all}\;\mathrm{BrdU}\;\mathrm{positive}\;\mathrm{cells}-\mathrm{labeled}\;\mathrm{cells}}{\mathrm{all}\;\mathrm{IdU}\;\mathrm{stained}\;\mathrm{cells}-\mathrm{labeled}\;\mathrm{cells}}$$
where “T_S_” is the length of the S-phase and “It” is the time period between BrdU and IdD labeling. From Equation (1), the Tc can be estimated.
(3)$$\mathrm{Tc}=\mathrm{Ts}\times \frac{\mathrm{all}\mathrm{cells}}{\mathrm{all}\mathrm{BrdU}\mathrm{stained}\mathrm{cells}-\mathrm{labeled}\mathrm{cells}}$$

3.This procedure has been proposed by Bialic et al. [70]. The principle of this method is to label S-phase cells with EdU, chase them with thymidine for varying time periods, and label the cells that are still in S-phase with BrdU. At regular intervals of time, from example at 0.5, 1.5, 3.5, 5.5 h, samples are collected. The rationale of this method is that as some cells tagged with EdU left the S-phase during the chase period, the proportion of cells co-labeled with both markers decreased with the prolongation of the chase length [6,70]. The time required for the earliest S-phase cells labeled with EdU and not tagged with BrdU corresponds to the duration of the TsS [70].

## 7. Conclusions

The detection of thymidine analogue incorporation into replicating DNA is a fast and useful technology to infer in an asynchronous population of cells, the fraction of S-phase cells, and to estimate cell-cycle time and the duration of the cycle phases in various biological contexts, including the normal development of cells, tissues, and organs, or the growth of a neoplasm. Moreover, the site, time of cell proliferation, and cellular kinetics of different kinds of cells have been estimated with several labeling schemes involving a single administration, continuous nucleotide analogue delivery, and double labeling with two thymidine analogues. It is important to indicate that before interpreting the results obtained with these labeled schemes, two important points should be considered. First, some of these approaches are not sensitive to the presence of quiescent or senescent cells. Second, the toxicity of the marker nucleosides and cell cycle progression. This is an important issue when a cumulative labeling sequence is required or two markers are administered. Therefore, the dose used of a given thymidine analogue is always a compromise. This is because it should provide a good signal and, in addition, it should not affect the proliferative dynamics of a cell.

## Figures and Tables

**Figure 1 biology-12-00885-f001:**
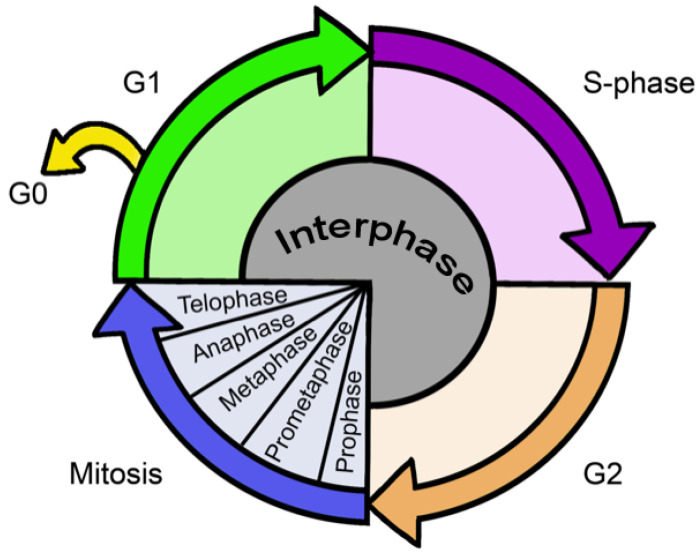
Schema of the cell cycle. The division cell cycle presents four discrete phases: G_1_, S, G_2_, and M phases (not to scale). G_0_ is the resting phase.

**Figure 2 biology-12-00885-f002:**
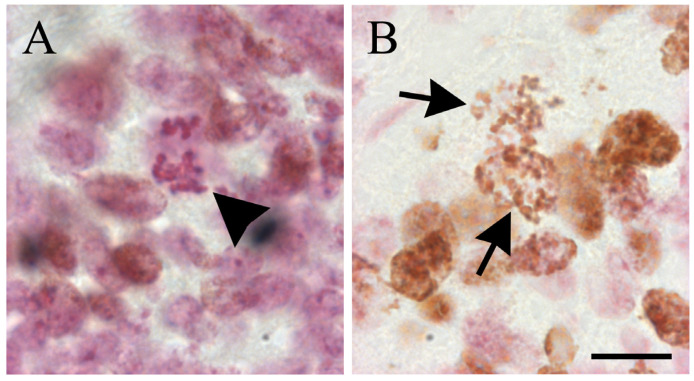
Examples of unlabeled (arrowhead in (**A**)) and labeled mitotic cells with BrdU in the cerebellar external granular layer of 8-day-old rats allowed to survive for 8 h after a single administration of BrdU (black arrows in (**B**)). Scale bar: 20 µm.

**Figure 3 biology-12-00885-f003:**
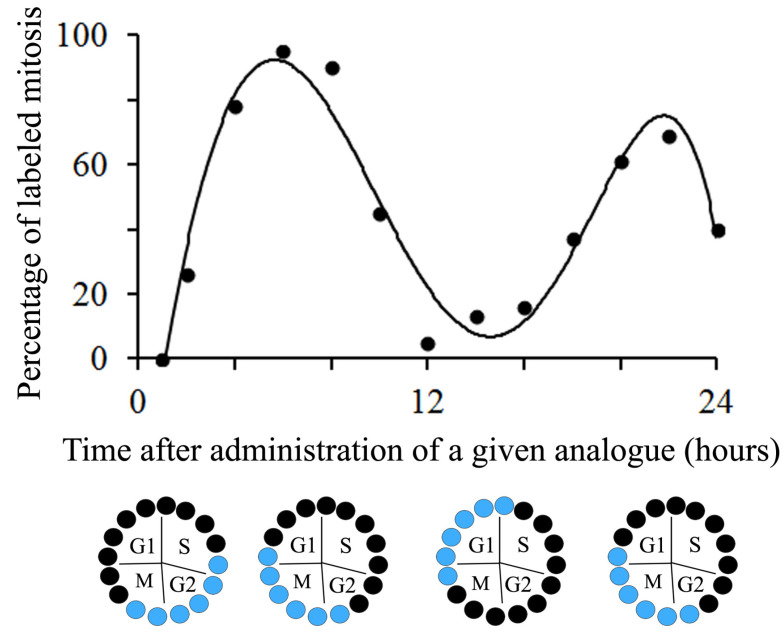
Plot of the percentage of labeled mitosis over time following a single administration of a given thymidine analogue. Black circles show unlabeled cells and blue circles indicate tagged cells.

**Figure 4 biology-12-00885-f004:**
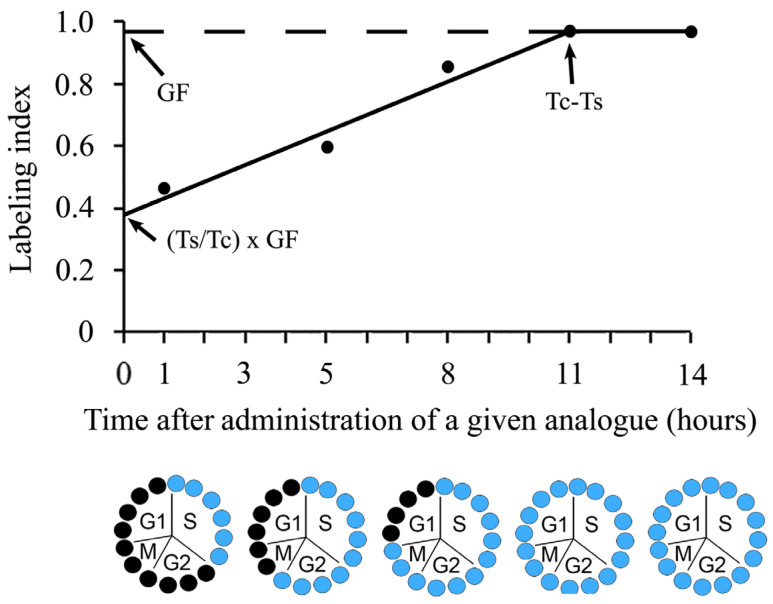
Changes in the labelling index throughout the time-course of a cumulative label with a given thymidine analogue. Black circles show unlabeled cells and blue circles indicate tagged cells. T_S_: the duration of the S-phase. Tc: the duration of the whole cycle. GF: the growth fraction.

**Table 1 biology-12-00885-t001:** Schedule of bromodeoxyuridine administration.

Survival Time after Bromodeoxyuridine (50 mg/Kg) Administration (Hours)						
0	0.5	1	1.5	2	2.5	3
A	S					
A		S				
A			S			
A				S		
A					S	
A						S

A: Administration; S: sacrifice. At least four animals should be used per each survival time.

**Table 2 biology-12-00885-t002:** Schedule of bromodeoxyuridine administration.

Survival Time after Bromodeoxyuridine (50 mg/kg) Administration (h)														
0	0.5	1	2	4	6	8	10	12	14	16	18	20	22	24
A	S													
A		S												
A			S											
A				S										
A					S									
A						S								
A							S							
A								S						
A									S					
A										S				
A											S			
A												S		
A													S	
A														S

A: Administration; S: sacrifice. At least four animals should be used per each survival time.

**Table 3 biology-12-00885-t003:** Schedule of bromodeoxyuridine administrations.

Survival Time after Bromodeoxyuridine (35 mg/kg) Administration of (hours)													
0	0.5	1	2	4	5	6	7	8	10	11	12	13	14
A		S											
A				A	S								
A				A			A	S					
A				A			A		A	S			
A				A			A		A			A	S

A: Administration; S: sacrifice. At least four animals should be used per each survival time.

## Data Availability

Not applicable.

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
