# Peer review of "Methods for Inferring Cell Cycle Parameters Using Thymidine Analogues"

_biology, 2023, doi:10.3390/biology12060885_

Round 1

Reviewer 1 Report

The review proposed in this manuscript aims to explain the value of using thymidine analogues to define the kinetics of cell cycle phases. The author of this review demonstrates real expertise in experimental approaches in this context and wishes to inform readers interested in these techniques of the limiting points and experimental bottlenecks they may encounter. The text is well-written and achieves its objectives.

Minor remarks

1. In Figure 1: G0 indicated in the caption is not present in the Schema of the cell cycle. The reviewer assumes the yellow arrow represents the exit into the resting state G0. This point should be clarified. In the same way, prometaphase is indicated in the text but not in the figure.

2. The abbreviations for the proportion of labelled mitotic cells are sometimes inverted or misspelt and indicated as PML, PLM, or FML. This point is confusing and needs to be corrected.

3. The equation TM=MI/100x TC/GF, the meaning of the abbreviation MI (Mitotic Index), is not given. Furthermore, Mitotic Index should be defined in the manuscript.

4. Black and blue circles (respectively unlabelled and tagged cells) under the plot of the percentage of labelled mitosis over time is easy to interpret in Figure 3 but far less understandable in Figure 2. According to this cell cycle drawing, the reviewer does not understand how the percentage of labelled mitosis may describe a sinusoidal curve. Can this point be clarified?

Author Response

Reviewer 1:

Thank you very much for your review of the present manuscript. Your constructive suggestions have been very valuable to me. The following points have been modified in accordance with your recommendations. I think that I have responded to the reviewer’s comment. Text modifications are detailed in blue.

The reviewer wrote (in bold, cursive writing):

Minor remarks

  1. In Figure 1: G0 indicated in the caption is not present in the Schema of the cell cycle. The reviewer assumes the yellow arrow represents the exit into the resting state G0. This point should be clarified. In the same way, prometaphase is indicated in the text but not in the figure.

I am sorry the first version was not clear enough. Following your suggestion, A new Figure 1 is provided. See page 2, lines 45-46.

  1. The abbreviations for the proportion of labelled mitotic cells are sometimes inverted or misspelt and indicated as PML, PLM, or FML. This point is confusing and needs to be corrected.

I appreciate this comment and completely agree. I am very sorry for my negligence at this point. Following your suggestion, the abbreviations for the proportion of labeled mitotic cells have been modified. See:

.- Page 7, line 236.

.- Page 8, line 247.

.- Page 9, line 268.

.- Page 9, line 269.

  1. The equation TM=MI/100x TC/GF, the meaning of the abbreviation MI (Mitotic Index), is not given. Furthermore, Mitotic Index should be defined in the manuscript.

I appreciate this comment and completely agree. I am very sorry for my negligence at this point. Following your suggestion, the meaning of the abbreviation MI (Mitotic Index) has been supplied. See:

.- Page 9, lines 286-287.

  1. Black and blue circles (respectively unlabelled and tagged cells) under the plot of the percentage of labelled mitosis over time is easy to interpret in Figure 3 but far less understandable in Figure 2. According to this cell cycle drawing, the reviewer does not understand how the percentage of labelled mitosis may describe a sinusoidal curve. Can this point be clarified?

I am sorry the first version was not clear enough. Following your suggestion, A new Figure (Figure 3) is provided (see page 9, line 263).

Once again, thank you very much for your comments and suggests.

Reviewer 2 Report

The author of this paper outlines various techniques that have been developed to estimate cell-cycle duration and the length of its phases. These methods include tritiated thymidine autoradiography and immunodetection of thymidine analogues after they have been incorporated into replicating DNA. By employing these approaches, valuable insights have been gained into the proliferative behavior of various cell types under different experimental conditions, using fixed tissue sections and either light or electron microscopy. Thymidine analog labeling has proven particularly beneficial in uncovering cell growth kinetics that would have been inaccessible through histological methods alone.

The manuscript presents captivating observations, exhibiting excellent writing and organization, along with an extensively updated bibliography. The figures demonstrate exceptional quality and provide comprehensive information. The conclusions are effectively presented and strongly backed by the data.

Author Response

Reviewer 2:

Thank you very much for your review of the present manuscript.

The author of this paper outlines various techniques that have been developed to estimate cell-cycle duration and the length of its phases. These methods include tritiated thymidine autoradiography and immunodetection of thymidine analogues after they have been incorporated into replicating DNA. By employing these approaches, valuable insights have been gained into the proliferative behavior of various cell types under different experimental conditions, using fixed tissue sections and either light or electron microscopy. Thymidine analog labeling has proven particularly beneficial in uncovering cell growth kinetics that would have been inaccessible through histological methods alone. 

The manuscript presents captivating observations, exhibiting excellent writing and organization, along with an extensively updated bibliography. The figures demonstrate exceptional quality and provide comprehensive information. The conclusions are effectively presented and strongly backed by the data.

Special thanks to you for your comments.

Reviewer 3 Report

The manuscript provides a comprehensive overview on the use of Thymidine analogues for cell cycle studies. Given the amount of literature that this manuscript covers, to make it more clear for the readers I would suggest expanding the text and adding more schematics to explain the experimental design described in different sections. Reproducing some data from cited papers, or at least schematically showing typical plots would help to make more sense of the text as well. Adding some tables describing different experimental conditions used in different publications (e.g., incubation times, thymidine analog concentrations, experimental systems used, purpose of the experiment) would make this review more useful as a methodological guidance.

Minor points:

- G0 is not labelled in Figure1

Line 295 and Fig 3 need to explain what GF, Tc, Ts are

The manuscript needs an extensive proofreading. Although overall the language is readable, there are many typos, wrong words and grammatical inconsistencies in the text. Examples are: "suturing" I instead of "saturating"  used multiple times in lines 169-173; "three example" mentioned but listed four in line 55; grammatically incorrect sentence in lines 53-55, in line 74 "BrdU, CldU and IdU need " should read as "BrdU, CldU and IdU detection needs", and many more similar examples.

Author Response

Reviewer 3:

Thank you very much for your review of the present manuscript. Your constructive suggestions have been very valuable to me. The following points have been modified in accordance with your recommendations. I think that I have now responded clearly to the reviewer’s comment. Text modifications are detailed in yellow.

The reviewer wrote (in bold, cursive writing):

The manuscript provides a comprehensive overview on the use of Thymidine analogues for cell cycle studies. Given the amount of literature that this manuscript covers, to make it more clear for the readers I would suggest expanding the text and adding more schematics to explain the experimental design described in different sections. Reproducing some data from cited papers, or at least schematically showing typical plots would help to make more sense of the text as well. Adding some tables describing different experimental conditions used in different publications (e.g., incubation times, thymidine analog concentrations, experimental systems used, purpose of the experiment) would make this review more useful as a methodological guidance.

To make this manuscript clearer for the reader than the previous version, three tables and a new figure are added.

See: page 5, lines 154-158 (Table 1).

See: page 7, lines 224-225 (Table 2).

See: page 8, line 241-244 (Figure 2).

See: page 10, lines 301-305 (Table3).

 Minor points:

- G0 is not labelled in Figure1

I am sorry the first version was not clear enough. Following your suggestion, a new Figure 1 is provided. See page 2, lines 45-48.

- Line 295 and Fig 3 need to explain what GF, Tc, Ts are

GF, Tc, Ts are previously explained in page 9, lines 276 to 289.

Following your suggestion, in figure legend for figure 3 Ts, Tc and GF are explained. See: page 11, line 322-324.

Comments on the Quality of English Language

The manuscript needs an extensive proofreading. Although overall the language is readable, there are many typos, wrong words and grammatical inconsistencies in the text. Examples are: "suturing" I instead of "saturating"  used multiple times in lines 169-173; "three example" mentioned but listed four in line 55; grammatically incorrect sentence in lines 53-55, in line 74 "BrdU, CldU and IdU need " should read as "BrdU, CldU and IdU detection needs", and many more similar examples.

I am grateful to you for this comment. I am very sorry for my negligence at this point. Following your suggestion, “suturing” has been replaced by “saturating”.

See: page 5, line 174.

See: page 5, line 178.

Following your suggestion, page 2, lines 53-55 has been corrected.

Following your suggestion, "BrdU, CldU and IdU need" has been replaced by “BrdU, CldU and IdU detection needs”.

See: page 3, line 74.

 I entirely endorse your concern. The text was revised by a native English speaker.

Special thanks to you for your good comments.

Round 2

Reviewer 3 Report

The manuscript clarity has been improved through the revision. 

Some mistakes have been corrected but the manuscript still needs a proofreading.